# Fractional Order Magnetic Resonance Fingerprinting in the Human Cerebral Cortex

**Viktor Vegh** [1,2,*] , **Shahrzad Moinian** [1,2] , **Qianqian Yang** [3] and **David C. Reutens** [1,2]

1   Centre for Advanced Imaging, University of Queensland, Brisbane 4072, Australia;
    s.moinian@uq.edu.au (S.M.); d.reutens@uq.edu.au (D.C.R.)
2   ARC Training Centre for Innovation in Biomedical Imaging Technology, University of Queensland,
    Brisbane 4072, Australia
3   School of Mathematical Sciences, Queensland University of Technology, Brisbane 4000, Australia;
    q.yang@qut.edu.au
*   Correspondence: v.vegh@uq.edu.au; Tel.: +61-7-3346-0363

**Abstract:** Mathematical models are becoming increasingly important in magnetic resonance imaging (MRI), as they provide a mechanistic approach for making a link between tissue microstructure and signals acquired using the medical imaging instrument. The Bloch equations, which describes spin and relaxation in a magnetic field, are a set of integer order differential equations with a solution exhibiting mono-exponential behaviour in time. Parameters of the model may be estimated using a non-linear solver or by creating a dictionary of model parameters from which MRI signals are simulated and then matched with experiment. We have previously shown the potential efficacy of a magnetic resonance fingerprinting (MRF) approach, i.e., dictionary matching based on the classical Bloch equations for parcellating the human cerebral cortex. However, this classical model is unable to describe in full the mm-scale MRI signal generated based on an heterogenous and complex tissue micro-environment. The time-fractional order Bloch equations have been shown to provide, as a function of time, a good fit of brain MRI signals. The time-fractional model has solutions in the form of Mittag–Leffler functions that generalise conventional exponential relaxation. Such functions have been shown by others to be useful for describing dielectric and viscoelastic relaxation in complex heterogeneous materials. Hence, we replaced the integer order Bloch equations with the previously reported time-fractional counterpart within the MRF framework and performed experiments to parcellate human gray matter, which consists of cortical brain tissue with different cyto-architecture at different spatial locations. Our findings suggest that the time-fractional order parameters, $\alpha$ and $\beta$, potentially associate with the effect of interareal architectonic variability, which hypothetically results in more accurate cortical parcellation.

**Keywords:** anomalous relaxation; magnetic resonance imaging; fractional calculus; cortical parcellation

## 1. Introduction

Magnetic resonance imaging (MRI) is a routinely used medical imaging modality known for its exquisite soft-tissue contrast. The power of this imaging modality arises from its sensitivity to changes in tissue composition, interpreted as changes in texture, intensity, shape and size of structures within images. Knowledge of the drivers behind the changes allows accurate diagnosis, treatment planning and monitoring of patients presenting with a range of diseases and disorders. In the brain, MRI-based image contrast is primarily due to microstructural differences in gray and white matter tissue. As such, assessment of diseases and disorders relates to how additional information appears in images (e.g., tumours) or how gray-white matter shape and size, topological features and spatial intensity vary with respect to the disease.

The current trend in MRI has been to supplement the qualitative image repertoire with quantitative, biologically relevant maps of tissue properties to improve diagnostic,

prognostic and treatment planning accuracy and specificity. This necessitates the research and development of analytical methods capable of inferring microstructural information from mm-scale measurements to bypass resolution limits imposed by the MRI hardware. The need for reproducible and quantitative MRI has prompted the development of methods capable of linking mm-scale MRI measurements with biologically interpretable information [1–7]. Here, an appropriate mathematical model in conjunction with specifically collected MRI data results in model parameters sensitive at the microscale. MRI voxel factors including myelin and iron content, tissue density, composition and microstructure orientation have been estimated in this manner, e.g., [1,3,8–10].

Classically, the Bloch equations describes how the MRI signal evolves over time as a function of spin–lattice ($T_1$) and spin–spin ($T_2$) relaxation. The equation holds for the case of an isotropic material, for example, a sample made up of water alone. However, when relaxation occurs in a complex tissue structure with tissue comprising of multiple constituents, the integer order differential equations describing sample magnetisation with a mono-exponential signal solution are no longer able to capture the trend in the signal as a function of time [11]. In this instance, the temporal MRI signal is said to deviate away from the expected mono-exponential trend. Models involving multiple exponentials (e.g., [12–14]) and time-fractional order derivative representations (e.g., [11,15]) have been proposed to better explain trends in MRI signals generated in real tissues. The reader is referred to a recent review article for a comprehensive overview of anomalous relaxation processes in MRI [16].

When using mathematical models applied to MRI data, the estimated model parameters are assumed to incorporate information on tissue microstructure. As such, a model needs to either exist or be developed and then fitted to specifically collected data. Estimation of model parameters involves a non-linear solver and, often, a good initial guess has to be made for the parameters to converge to realistic values [11]. This approach has shortcomings, including the potential for overfitting and making sure the non-linear solver converges to a realistic solution. A different approach is to generate a so-called dictionary based on discrete parameter choices. For each set of model parameters, a simulated signal based on the Bloch equations can be generated and matched with the observed signal. This latter framework is referred to as magnetic resonance fingerprinting (MRF) [17] and has become a routine tool in MRI. The parameters associated with the best matched signal are said to be those most reflective of the tissue bulk under investigation. Depending on how parameters are discretised, MRF dictionaries can become very large and the computational burden of matching the signal to the dictionary expands. Irrespective of the approach taken to estimate model parameters, the parameters are depicted as spatially resolved maps (i.e., model is applied at each pixel/voxel location), which are quantitative images.

MRF signals have mostly been generated using the classical Bloch equations and other approaches not involving fractional calculus [18,19]. Recently, the use of the time-fractional Bloch equations was investigated [20]. The MRF approach provides a platform for exchanging models with an associated adjustment of the parameter space. Building on previous work on anomalous relaxation in MRI [11], our aim here was to investigate further the link between the time-fractional order parameters and the expected variation in tissue microstructure. The overall motivation being that accurate delineation of microstructurally distinct regions of the human cerebral cortex in individuals, i.e., known as cortical parcellation, is fundamental for understanding structure–function relationships in the brain [21]. Additionally, specific variations in model parameters may shed light on aging and how different diseases affect cortical regions [22,23] and improve delineation of abnormal tissue in the surgical setting [24,25]. Current methods in the cortical parcellation area mostly involve the assessment of changes in MRI relaxation times (such as $T_1$ and $T_2$) [26–29] and it is becoming increasingly evident that these model parameters are unlikely capable of distinguishing interareal structural variations throughout the whole human cerebral cortex [30–33]. In what follows, we outline our approach of using the time-fractional Bloch equation in the MRF context for parcellating the cerebral cortex in individuals.

## 2. Materials and Methods

### 2.1. Bloch Equations

In this subsection, we provide the formulation used in the study based on a previously published model [11]. We first describe the general time-fractional order Bloch model, which can be simplified to the classical case. The equations describing the evolution of magnetisation within a spin system and then related to an MRI signal.

### 2.1.1. Time-Fractional Order Model

We consider the case of spin–spin relaxation in an inhomogeneous magnetic field due to microscale variations in tissue microstructure. In the rotating frame a residual frequency offset presents [11] $\Delta\omega = 2\pi\Delta f$, where $\Delta f$ is in units of Hz and $T_2$ is replaced by $T_2^*$ and, strictly, relaxation times satisfy $T_1 \geq T_2 \geq T_2^*$ and measured in units of s. Whilst different approaches may be considered to arrive at a time-fractional order Bloch equation [15], here, the approach of fractionalising the time derivative was taken:

$$
\begin{aligned}
\tau_1^{\alpha-1}{}_0^C D_t^\alpha M_z(t) &= \frac{M_0 - M_z(t)}{T_1}, \\
\tau_2^{\beta-1}{}_0^C D_t^\beta M_x(t) &= -\frac{1}{T_2^*}M_x(t) + \Delta\omega M_y(t), \\
\tau_2^{\beta-1}{}_0^C D_t^\beta M_y(t) &= -\frac{1}{T_2^*}M_y(t) - \Delta\omega M_x(t),
\end{aligned}
\tag{1}
$$

where constants $\tau_1$ and $\tau_2$ are incorporated to preserve units and $M_0$ in units of A/m is the net magnetisation produced by the MRI scanner. We should note that the z-component magnetisation, $M_z(t)$, recovers in time with time constant $T_1$ and transverse magnetisations, $M_x(t)$ and $M_y(t)$, decay in time with time constant $T_2^*$. The time fractional derivative follows the Caputo (${}_0^C D^\gamma$) definition [34] of fractional order $\gamma$ described in the following:

$$
{}_0^C D_t^\gamma f(t) = \frac{1}{\Gamma(1-\gamma)}\int_0^t \frac{f'(\lambda)}{(t-\lambda)^\gamma}d\lambda, \; 0 < \gamma \leq 1,
\tag{2}
$$

where $\Gamma$ denotes the Gamma function. The Caputo fractional derivative is the appropriate choice because it obeys the correct physical meaning for the initial condition; in particular, the Laplace transform of this operator involves the true observed initial magnetization. This is not the case for the Riemann–Liouville fractional derivative, which instead involvs a non-physical fractional initial condition (refer to Equation (3.6) in [35]). The solution to (1), as described in [11], takes the following form:

$$
M_z(t) = M_z(0)E_\alpha\left(-\frac{\tau_1^{1-\alpha}t^\alpha}{T_1}\right) + \frac{M_0}{T_1}\tau_1^{1-\alpha}t^\alpha E_{\alpha,\alpha+1}\left(-\frac{\tau_1^{1-\alpha}t^\alpha}{T_1}\right),
\tag{3}
$$

$$
M_x(t) = \frac{M_x(0)-iM_y(0)}{2}E_\beta\left(-\frac{\tau_2^{1-\beta}t^\beta}{T_2^*}+i\Delta\omega\tau_2^{1-\beta}t^\beta\right) + \frac{M_x(0)+iM_y(0)}{2}E_\beta\left(-\frac{\tau_2^{1-\beta}t^\beta}{T_2^*}-i\Delta\omega\tau_2^{1-\beta}t^\beta\right)
$$

$$
M_y(t) = \frac{M_y(0)+iM_x(0)}{2}E_\beta\left(-\frac{\tau_2^{1-\beta}t^\beta}{T_2^*}+i\Delta\omega\tau_2^{1-\beta}t^\beta\right) + \frac{M_y(0)-iM_x(0)}{2}E_\beta\left(-\frac{\tau_2^{1-\beta}t^\beta}{T_2^*}-i\Delta\omega\tau_2^{1-\beta}t^\beta\right)
$$

where $E_{a,b}(z) = \sum\limits_{k=0}^{\infty} z^k/\Gamma(ak+b)$ is the two-parameter Mittag–Leffler function and by definition $E_{a,1}(z) \equiv E_a(z)$ and $E_1(z) = e^z$. The computational complexity and long memory effect are pertinent challenges when numerical solutions to fractional differential equations are sought. The model for the MRI fingerprinting we consider in this paper is a time-fractional differential equation with an analytical solution as shown in (3), hence numerical methods are not involved. Nevertheless, the analytical solution does involve the evaluation of the Mittag–Leffler function, achieved through a standard implementation using the optimal parabolic contour (OPC) algorithm described in [36] and the MATLAB code is available at MATLAB Central File Exchange #48154. Noticeably, $M_x(t)$ and $M_y(t)$ involve a different time-fractional order than $M_z(t)$. This is because, from a physics

perspective, different processes drive magnetisation change for these components (i.e., $T_1$ versus $T_2^*$ effects). Since $T_1 \gg T_2^*$, the time scale over which $M_z(t)$ recovers to $M_0$ is also quite different to the time scale over which $M_x(t)$ and $M_y(t)$ tend to zero. This becomes important later in the context of MRI data collection.

The condition $T_1 \geq T_2 \geq T_2^*$ holds in the case of the integer order model. In the time fractional case, we can show that $t^\beta \tau_2^{1-\beta}/T_2 \geq t^\alpha \tau_1^{1-\alpha}/T_1$ has to be met. Refer to Appendix A for the derivation of this condition.

### 2.1.2. Integer Order Model

In (3) we may set $\alpha = \beta = 1$, which then results in the following equations.

$$
\begin{aligned}
M_z(t) &= M_z(0)e^{-\frac{t}{T_1}} + M_0\left(1 - e^{-\frac{t}{T_1}}\right), \\
M_x(t) &= \frac{M_x(0) - iM_y(0)}{2}e^{-\frac{t}{T_2^*} + i\Delta\omega t} + \frac{M_x(0) + iM_y(0)}{2}e^{-\frac{t}{T_2^*} - i\Delta\omega t}, \\
M_y(t) &= \frac{M_y(0) + iM_x(0)}{2}e^{-\frac{t}{T_2^*} + i\Delta\omega t} + \frac{M_y(0) - iM_x(0)}{2}e^{-\frac{t}{T_2^*} - i\Delta\omega t}.
\end{aligned}
\tag{4}
$$

It is possible to additionally set $\Delta\omega = 0$ in (3), which would then be the case of spin–spin relaxation in the absence of field inhomogeneities. Note, according to the Larmor equation (i.e., $\Delta f = 42.578 \times 10^6 \Delta B$ for hydrogen spins where $\Delta B$ is the induced change in magnetic flux density), an induced change in field results in an induced shift in frequency. During MRI data acquisition, it is possible to apply a so-called radio frequency refocusing pulse which ensures signals are not de-phased at the time of signal acquisition. With the use of such an approach, $T_2^*$ in (4) would have to be exchanged by $T_2$ and $\Delta\omega = 0$ can be assumed. The reduced form of time-fractional Bloch equations has been used previously as well [37,38].

### 2.1.3. From Magnetisation to an MRI Signal

The spin system considered in MRI results in a net magnetisation vector which can be considered to undergo precession in each MRI voxel known as a volume pixel. Since the reference magnetisation, $M_0$, is defined in the z-coordinate direction, the precession of spins occurs in the plane perpendicular to the z-direction. That is, in the reference frame, precessional frequency exists for $M_x(t)$ and $M_y(t)$, but not for $M_z(t)$ (see in (3) that a frequency shift is only present on $M_x(t)$ and $M_y(t)$ and for additional information refer to [11]). This means the signal detection system, which is sensitive to a specific frequency band, produces a signal due to $M_x(t)$ and $M_y(t)$ alone. Traditionally, the induced signals in the induction coils used to collect the MRI signal are converted to a magnitude signal described in the following:

$$
S(t) = \sqrt{M_x(t)^2 + M_y(t)^2}
\tag{5}
$$

which after the subsitution of (3) conveniently becomes:

$$
S(t) = S_0\sqrt{E_\beta\left(-t^\beta\left(\frac{1}{T_2^*} + i\Delta\omega\right)\right)E_\beta\left(-t^\beta\left(\frac{1}{T_2^*} - i\Delta\omega\right)\right)} = S_0\left|E_\beta\left(-t^\beta\left(\frac{1}{T_2^*} + i\Delta\omega\right)\right)\right|,
\tag{6}
$$

where $S_0 = \sqrt{M_x^2(0) + M_y^2(0)}$ is the signal when $t = 0$. After careful inspection, we find that (6) is insensitive to $M_z(t)$, suggesting the z-component of magentisation behaves independently of $M_x(t)$ and $M_y(t)$. However, this is not strictly true. In an MRI system, magnetisation at $t = 0$ is preserved, meaning $M_0 = \sqrt{M_x^2(0) + M_y^2(0) + M_z^2(0)}$. Rearranging this relationship results in two definitions for $S_0$, namely as in (6) and additionally $S_0 = \sqrt{M_0^2 - M_z^2(0)}$. In the MRF context, the $S_0$ term involving $M_0$ and $M_z(0)$ ensures that the estimation of all model parameters can be approximated by matching (6) with the experimentally acquired MRI signal. We should point out that in the $T_2^*$ regime, the temporal magnetisation vector is complex valued, implying that a phase is also generated

during MRI data acquisition, which takes the value $\Theta = \Delta\omega \times t$ based on (4). This phase can be interpreted as a shift in frequency arising due to microscale variations in tissue magnetic properties inducing small magnetic field variations. Note, according to the Larmor equation, $\Theta = 2\pi \times 42.578 \times 10^6 \Delta B \times t$, phase increases with $t$ and $\Delta B$. We should additionally point out that $M_z(t)$ is real valued and it is not influenced by $\Delta B$. Hence, one may expect static field inhomogeneities introduced by the sample within MRI voxels to only influence $M_x(t)$ and $M_y(t)$ and not $M_z(t)$.

### 2.2. Parameter Estimation Using MRF

Suppose the MRI instrument collects signals, $S(t)$. Then one approach of generating model parameters would be to fit the signal model, described by (3) and (5), to the acquired MRI temporal signal, $S(t)$. This approach requires a non-linear fitting algorithm with good initial parameter guesses to be able to achieve reasonable parameter estimates, especially when MRI data contains noise, as is normally the case [11].

In MRF a dictionary matching approach to parameter estimation is considered. Suppose the parameter space is known and it can be represented using discrete values. The MRF dictionary is often represented as a matrix, where each column defines a specific parameter and each row is a unique set of parameters. The key to each dictionary entry is the unique MRF signal, $S_{MRF}(t)$, which can then be simulated using (3) and (5) for each parameter combination [17]. The best matched $S_{MRF}(t)$ to $S(t)$ is considered to have the parameters which best parameterise the voxel. After performing the matching for each voxel, where the metric to match signals is commonly the dot product between the time series signals, i.e., $\max_{P} S_{MRF}(t) \cdot S(t)$ over parameters P, the resultant parameters are depicted as spatially resolved maps, which are sometimes referred to as quantitative images. Based on (3), each voxel can be parameterised in terms of $T_1$, $T_2^*$ and $\Delta\omega$. Time fractional exponents $\alpha$ and $\beta$ are assumed to result in additional parametric maps containing information relating to tissue microstructure and constituents. For convenience, $\tau_1$ and $\tau_2$ are assumed to equal 1.

The first step in generating the MRF dictionary is to decide on the practical bounds for each of the parameters, which are often constrained by existing knowledge. For example, it is known that gray and white matter $T_1$ values in the brain range between 500 ms to 3000 ms and similarly bounds can be placed on $T_2^*$ and $\Delta\omega$. The fractional exponents by definition have to satisfy $\alpha, \beta \leq 1$ and typically they are close to 1. Here, we opted to create the MRF dictionary using the following parameter ranges.

$$\{500 \leq T_1 \leq 3000 \mid T_1 = 500, 520, \ldots, 2000, 2030, \ldots, 2700, 2760, \ldots, 3000 \text{ ms}\},$$
$$\{14 \leq T_2^* \leq 49 \mid 14, 16, \ldots, 40, 43, \ldots, 49 \text{ ms}\},$$
$$\{0 \leq \Delta f \leq 45 \mid \Delta f = 0, 5, \ldots, 45 \text{ Hz}\}$$
$$\{0 < \alpha, \beta \leq 1 \mid \alpha, \beta = 0.6, 0.7, 0.8, 0.85, 0.9, 0.95, 1\}$$

One may consider the MRF signal matching approach as a constrained parameter estimation problem. The MRF dictionary is essentially a discrete representation of the parameter space defined through certain combinations of individual parameters. The number of parameter sets in the dictionary can rapidly become very large, as governed by the increments between parameters. Furthermore, the creation of the discrete parameter space from which signals are simulated is a combinatorial problem, wherein physical limits can be adopted to reduce MRF dictionary size.

### 2.3. Relating MRI Data to the Bloch Model

The flip angle ($\theta$ for simplicity) used in the MRI data acquisition equation 'flips' the $(0, 0, M_0)^T$ magnetisation vector by an angle $\theta$ with respect to the $z$-axis. This, when applied at $t = 0$, results in a new z-component magnetisation, such that $M_z(0) = M_0 \cos\theta$. Substitution of this value into $S_0 = \sqrt{M_0^2 - M_z^2(0)}$ yields $S_0 = M_0 \sin\theta$, which is the term governing the MRI signal amplitude in (6). After the application of an initial flip of the

magnetisation vector, the signal is collected at $t = TE$. However, the next application of the flip angle occurs at $t = TR$, at which point, according to (4), left over $z$-component magnetisation is $M_z(TR)$. This value becomes the $M_z(0)$ for the next TR cycle or MRF repetition. In this manner, information on $T_1$ is encoded into $S_0$. Note, changes in flip angle results in a non-linear modulation of the MRI signal, essentially bringing more complexity into the MRF signal repetitions. The matching of the simulated and acquired signals (as a function of MRF repetitions over TRs) results in a signal which best matches the measured MRI signal. The parameters used to generate that simulated signal are then used to create parametric maps in a pixel-by-pixel manner.

*2.4. MRI Data Collection*

Six mixed gender volunteers between 27 and 35 years of age participated in the study. The MRI sequence was implemented and used to acquire data using the 7T whole-body MRI research scanner (Siemens Healthcare, Erlangen, Germany) located at the Centre for Advanced Imaging, University of Queensland, Brisbane, Australia. Data acquisition involved a 3D echo planar imaging (EPI) MRF sequence [39] with 1000 frames made up of 3D MRF images with the following parameters: repetition time (TR) = 41–99 ms; flip angle (FA) = 10–77°; echo time (TE) = 12–48 ms; partial Fourier phase = 6/8; voxel size = 1.4 × 1.4 × 1.4 mm; and matrix size = 142 × 142 × 88. We used GRAPPA parallel imaging [40] in both phase encoding (with acceleration factor = 3 and reference lines = 36) and slice encoding directions (with acceleration factor = 2 and reference lines = 12). Note, the dot product based dictionary matching is robust with undersampled data (i.e., with the use of GRAPPA). Chemical-shift-selective (CHESS) fat saturation technique [41] was used to reduce common artefacts observed in EPI sequences at high field scanners [42]. The sinusoidal FA pattern used for the MRF acquisitions, shown in Figure 1a, was assumed from [43]. Abrupt FA changes at the final MRF frames were added for increased sensitivity of the MRF signals to transmit field variations, as suggested previously [44].

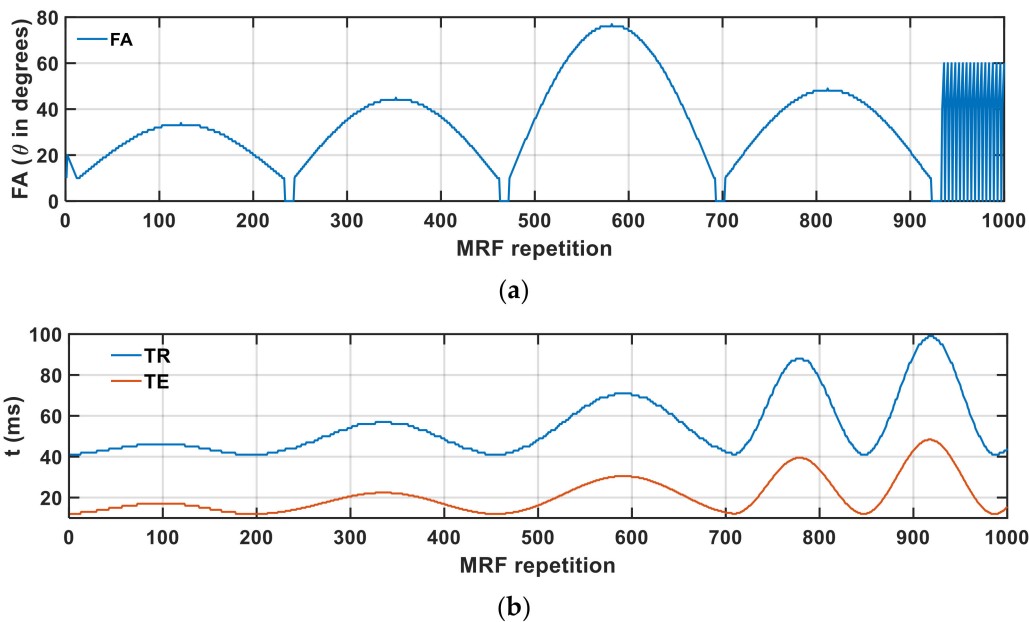

**Figure 1.** Shown are pseudo-randomised pattern of (**a**) flip angles (FA) and (**b**) the repetition time (TR) in blue and echo time (TE) in red used to acquire the 3D MRI data for each MRF repetition.

We additionally used pseudo-randomised patterns of TE variation suggested by Rieger et al. [45], see Figure 1b, to improve signal-to-noise ratio as a result of the large number of small TE values. The use of small TE values also resulted in a reduction in total acquisition time, as TR (the sum of which scales with the total data acquisition time) can be made proportionally small with respect to TE. The alternating TE pattern has also been shown to increase sensitivity of MRF signals to $T_1$ and $T_2^*$ variations [45]; the positive outcome of which is the improved estimation of these values.

### 2.5. Model Selection and Estimation Error

Traditionally the integer order Bloch equations (i.e., (4)) have been used to generate simulated MRF signals based on predefined sets of discrete parameters. By replacing the classical Bloch equations with the time-fractional counterpart (i.e., (3)), the number of model parameters increases by three, namely $\alpha$, $\beta$ and $\Delta f$, assuming $\tau_1 = \tau_2 = 1$. According to the Akaike information criteria [46], for example, the residual of the fit would have to be penalised due to an increase in the number of model parameters. As such, the residual error of fit using the time-fractional model is expected to be smaller than using the classical approach and a penalty associated with increased number of model parameters has to be considered. Nonetheless, we are particularly interested not only on whether the time-fractional Bloch model is able to better fit the data but also if the additional model parameters contain new information. The ability to provide information on cyto-architecturally different brain regions based the time-fractional Bloch equations could suggest that time fractional order plays a key role in MRI signal formation.

### 2.6. Human Brain Cortical Parcellation

Based on cyto-architectonic and myelo-architectonic knowledge in the brain, eleven cortical areas from the Jülich histological atlas of the human brain were selected [47]. These included the primary somatosensory cortex (BA1, BA2, BA3a and BA3b), primary motor cortex (BA4a and BA4p), premotor cortex (BA6), primary and secondary visual cortex V1 (BA17) and V2 (BA18) and the Broca areas BA44 and BA45. These cortical areas are known to have microstructurally distinct histological features [48–53].

## 3. Results

We provide MRF simulation results based on the time-fractional Bloch equations, followed by results on the separability of different cortical regions in the human brain based on time fraction Bloch model parameters.

### 3.1. Expected Changes in the MRF Signal

The MRF signal follows a pseudo-random pattern as defined by the acquisition protocol flip angle (i.e., $\theta$), TE and TR. The plots provided in Figure 2 show how the MRF signal based on the acquisition protocol parameters provided in Figure 1 evolve over MRF acquisition repetitions. The signal is acquired at $t = \text{TE}$ and $M_z(t)$ is allowed to evolve until t = TR; the sequence is then repeated. The different plots consider changes in $T_1$, $T_2^*$, $\alpha$ and $\beta$ and in each case a freqeuncy shift of $\Delta f = 10$ Hz. Distinct changes in $T_1$ (Figure 2e) and $T_2^*$ (Figure 2f) result in specific changes in the MRF signal over repetitions. We may also notice that a change in $\alpha$ (Figure 2b) results in a change in $M_z(t = TR)$, essentially modulating the amount of magnetisation available for each repetition. The parameter $\beta$ acts on the transverse magnetisation and thus on $S(t)$, which is the transverse magnetisation derived MRF signal (Figure 2c). Changes in both $T_2^*$ and $\Delta f$ appear to amplitude modulate $S(t = TE)$.

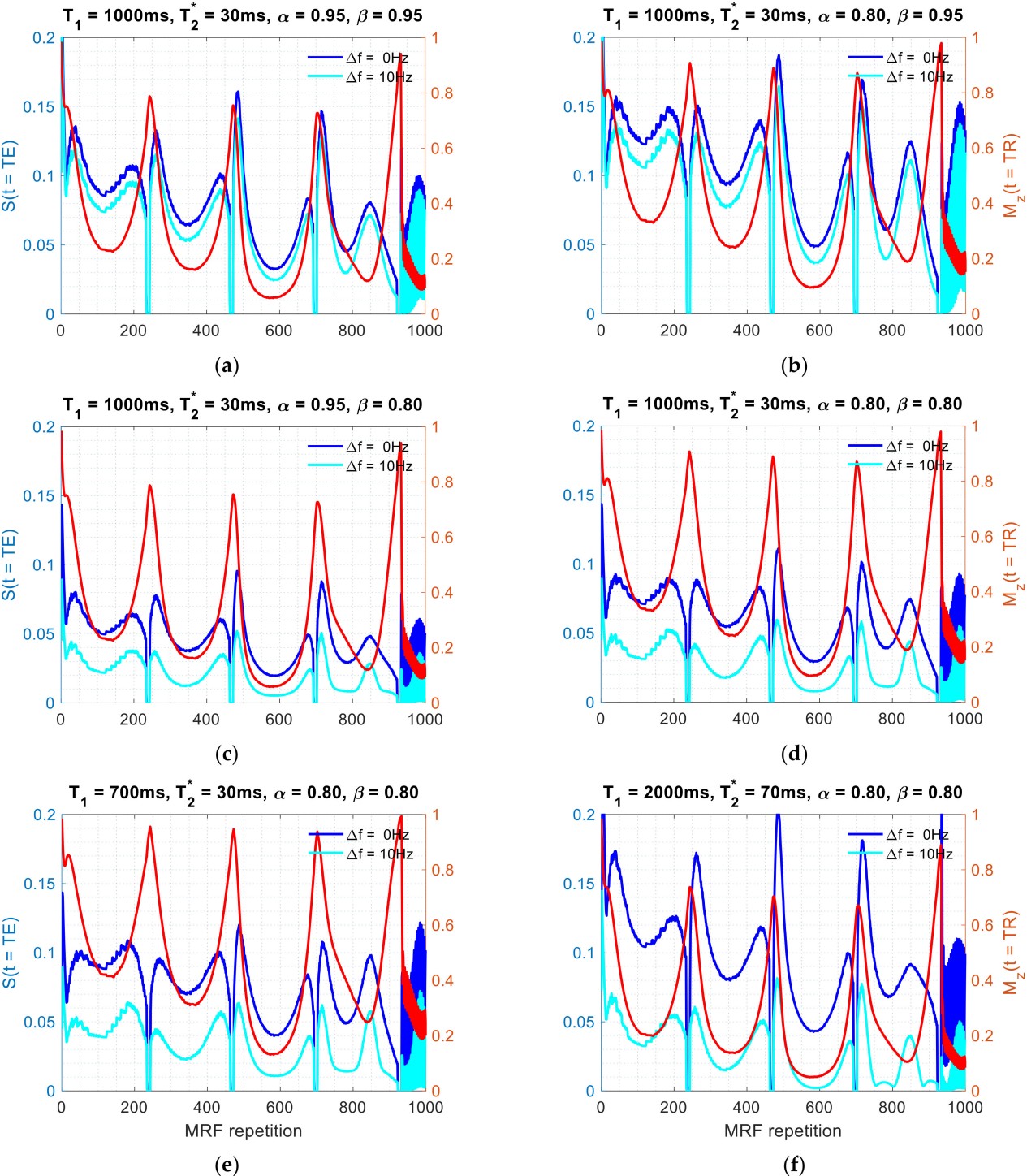

**Figure 2.** The MRF signal (i.e., $S(t)$ in blue) and spin–lattice magnetisation (i.e., $M_z$(t) in red) evolves as a function of MRF repetition. The MRF signal is acquired at $t$ = TE and magnetisation evolves until $t$ = TR before the sequence is repeated for a new flip angle (i.e., $\theta$) and TE and TR. $T_1$ and $T_2^*$ values have been chosen in view of representative values in the human brain. Displayed are time fractional Bloch equations simulations when (**a**) $\alpha$ and $\beta$ are close to the integer order case and effects of changing (**b**) $\alpha$, (**c**) $\beta$, (**d**) both $\alpha$ and $\beta$, (**e**) $T_1$ and (**f**) both $T_1$ and $T_2^*$ are depicted. In each case, a distinct change in $\Delta f$ has been plotted as well.

### 3.2. Time-Fractional Bloch Model Parameter Sensitivity

Time fractional Bloch equations simulations were performed based on the parameter ranges provided in Section 2.2. For each simulation, i.e., a data point, random values for $T_1$, $T_2^*$, $\alpha$, $\beta$ and $\Delta f$ were chosen from their respective intervals. Additionally, a random change was applied to each parameter and assessed in a one-by-one manner and, for the sake of this analysis, repeated 500 times (i.e., 500 data points were generated in each case). The error applied to each parameter was limited to a maximum of 25% with respect to the actual parameter value. This process resulted in the generation of parameter sensitivity maps in view of the angle produced by the dot product matching, as shown in Figure 3.

It appears that from the results provided, the time-fractional Bloch model, which includes $T_1$, $T_2^*$, $\alpha$, $\beta$ and $\Delta f$, overfits the MRF signal; observe the wide ranging values for $T_2^*$, $\beta$ and $\Delta f$. Values for $T_1$ and $\alpha$ appear not to be overfitted and good sensitivity to these parameters can be achieved. Observe the slope on the fit and also the clustering of data points about the linear regression line.

Noting that $T_2^*$ is a physical parameter and findings from Figure 3 suggest model degeneracy due to an interplay between $\beta$ and $\Delta f$, these are indicative of results shown in Figure 2 wherein both of these parameters' amplitude modulate $S(t = TE)$. To overcome the overfitting problem, we may consider two options: set $\beta = 1$ or $\Delta f = 0$. The former choice causes a fundamental problem, since the substitution of $\beta = 1$ into (6) results in a signal insensitive to changes in $\Delta f$. As such, we opted to investigate the option of setting $\Delta f = 0$ and leaving $\beta$ as a free model parameter. In Figure 4 parameter sensitivity results when $\Delta f = 0$ are provided. The results suggest that parameter insensitivities concluded from Figure 3 can be overcome by setting $\Delta f = 0$. Judging from the regression line slope, the best sensitivity is achieved for $\alpha$, followed by $\beta$ and $T_1$ and lastly $T_2^*$. These results are in view of the paradigm described in Figure 1 and may not be generalisable across different MRF acquisition protocols wherein different TEs and TRs may be set.

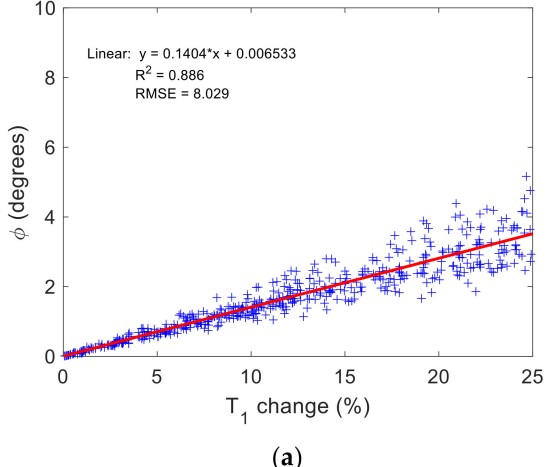

(a)

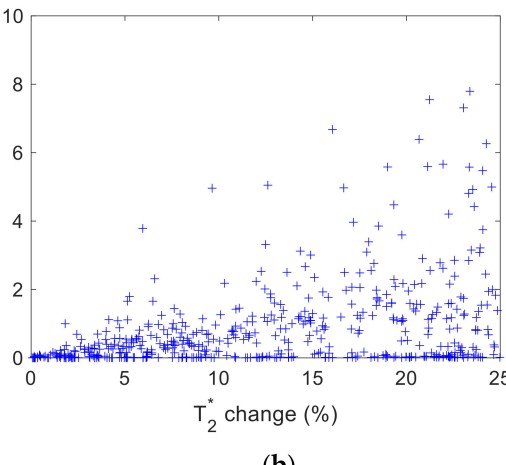

(b)

**Figure 3.** *Cont.*

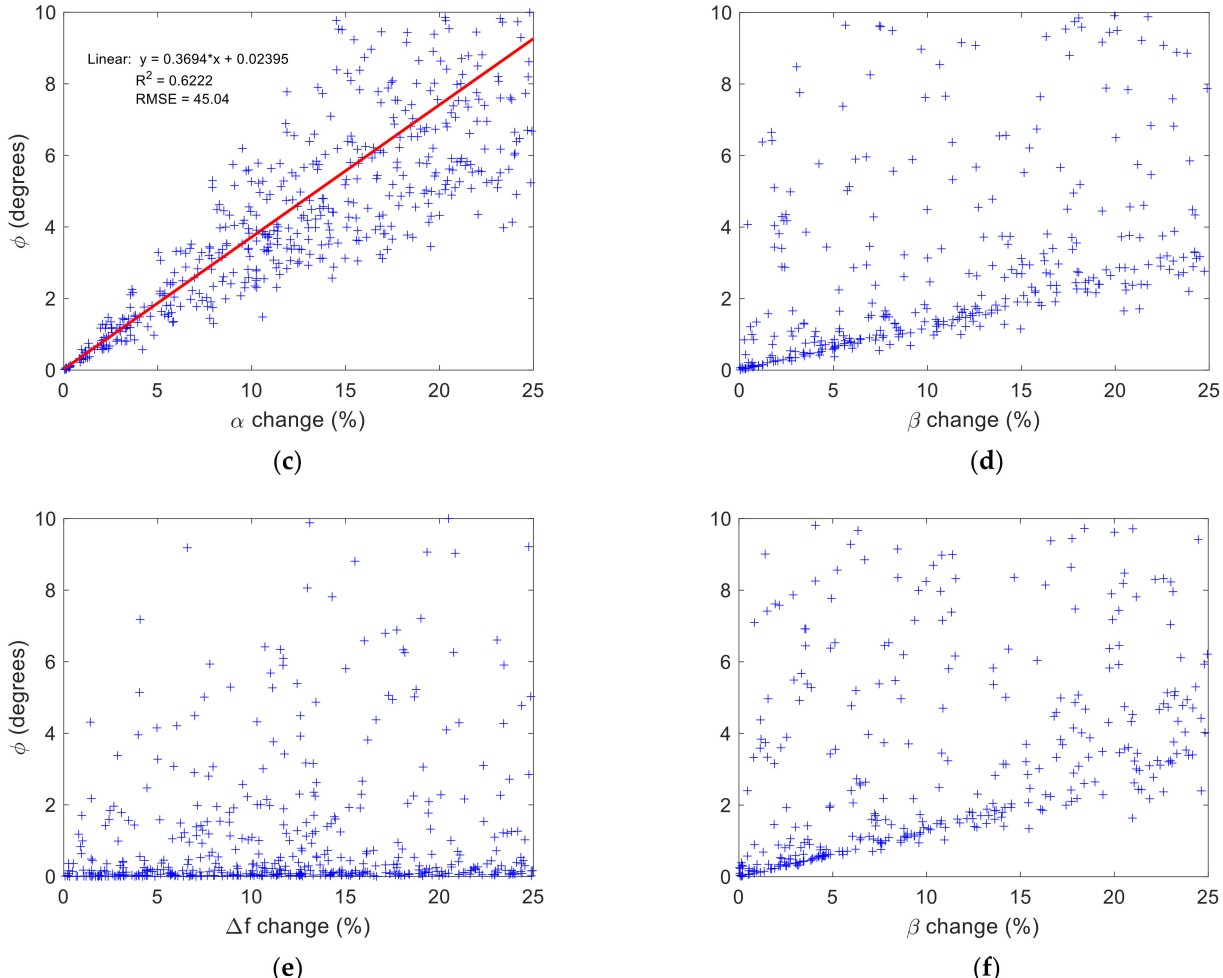

**Figure 3.** Illustration of the sensitivity of time fractional Bloch model to changes in each of the model parameters. On the *x*-axis, the change in the parameter is provided, plotted against the actual change in the signal and measured as the angle between the two vectors (i.e., dot product, as used for matching the simulated signal with acquired MRF signal). Each data point on each plot has been generated by randomly choosing parameter values from the appropriate parameter ranges. Each figure contains 500 random instances and percentage error change in the parameter was also chosen random in the range (0, 25%). Displayed are sensitivities to changes in (**a**) $T_1$, (**b**) $T_2^*$, (**c**) $\alpha$, (**d**) $\beta$, (**e**) $\Delta f$ and (**f**) assuming the special case of $\alpha = 1$ for $\beta$ sensitivity. For each plot, a linear regression was performed and $R^2$ and root-mean-squared error (RMSE) are provided.

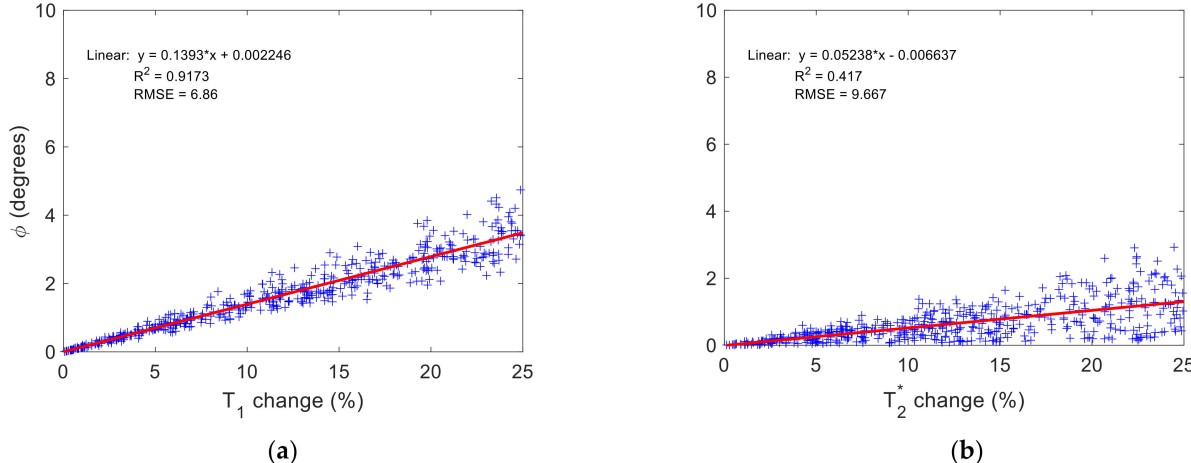

**Figure 4.** *Cont.*

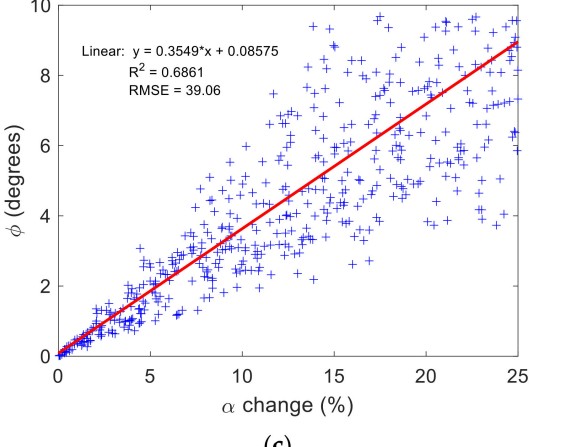
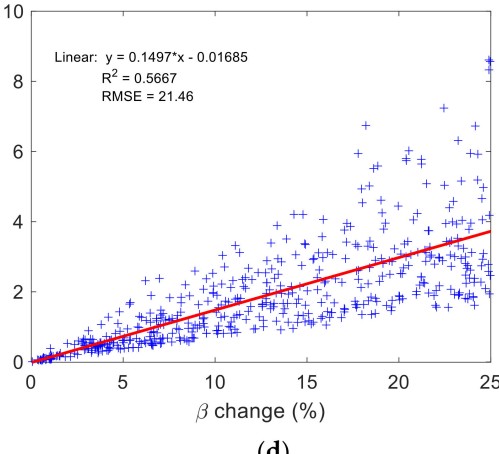

(**c**)　　　　　　　　　　　　　　　　　　(**d**)

**Figure 4.** Illustration of the sensitivity of time fractional Bloch model to changes in each of the model parameters under the assumption $\Delta f = 0$. On the *x*-axis, the change in the parameter is provided, plotted against the actual change in the signal and measured as the angle between the two vectors (i.e., dot product, as used for matching the simulated signal with acquired MRF signal). Each data point on each plot has been generated by randomly choosing parameter values from the appropriate parameter ranges. Each figure was repeated for 500 random instances and percentage error change in the parameter was also chosen random in the range (0, 25%). Shown are sensitivities to changes in (**a**) $T_1$, (**b**) $T_2^*$, (**c**) $\alpha$ and (**d**) $\beta$. For each plot, a linear regression was performed and R2 and root-mean-squared error (RMSE) are provided.

### 3.3. Parameter Selectivity to Different Cortical Regions in the Human Brain

For the purpose of being able to appreciate how the data were acquired, Figure 5 illustrates an MRF image slice in the human brain and MRI signal evolution over MRF repetitions. At each location in the brain in the image, the MRF matching based on the dot product metric was applied and parameters associated with the best matched signal were deemed to reflect location specific tissue parameters (i.e., $T_1$, $T_2^*$, $\alpha$ and $\beta$). Figures 6 and 7 illustrate the spatially resolved maps for each of the time-fractional Bloch equations parameters for an example slice at different locations for two of the participants.

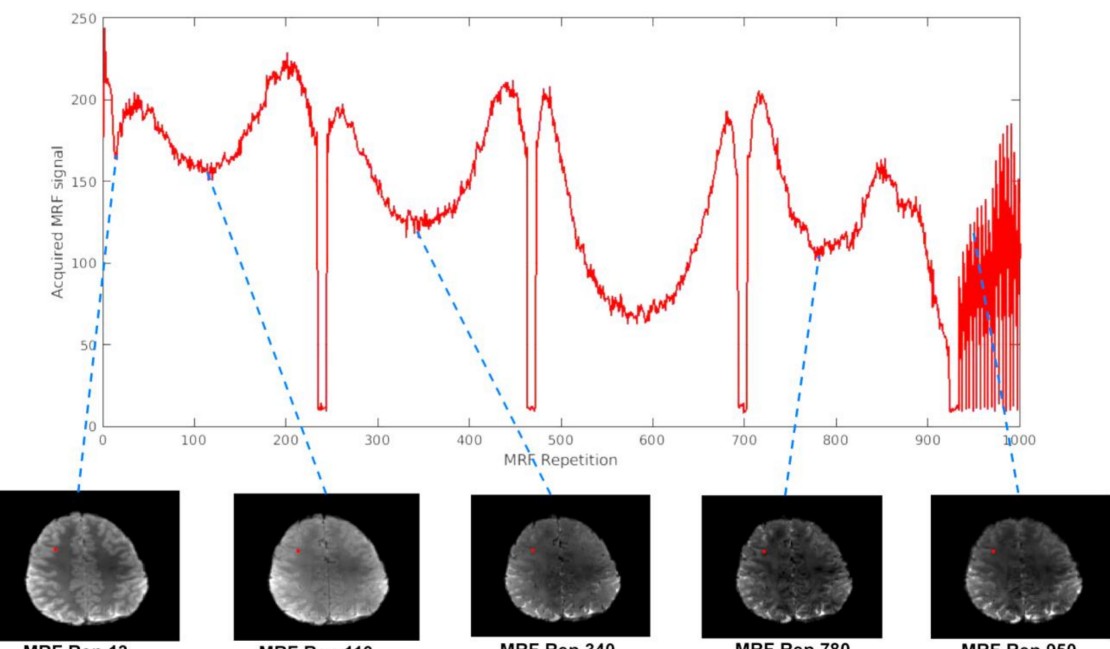

**Figure 5.** An illustration of the MRF data acquisition and how the signal may evolve at a particular location within the image. Example images at certain MRF repetitions have been provided.

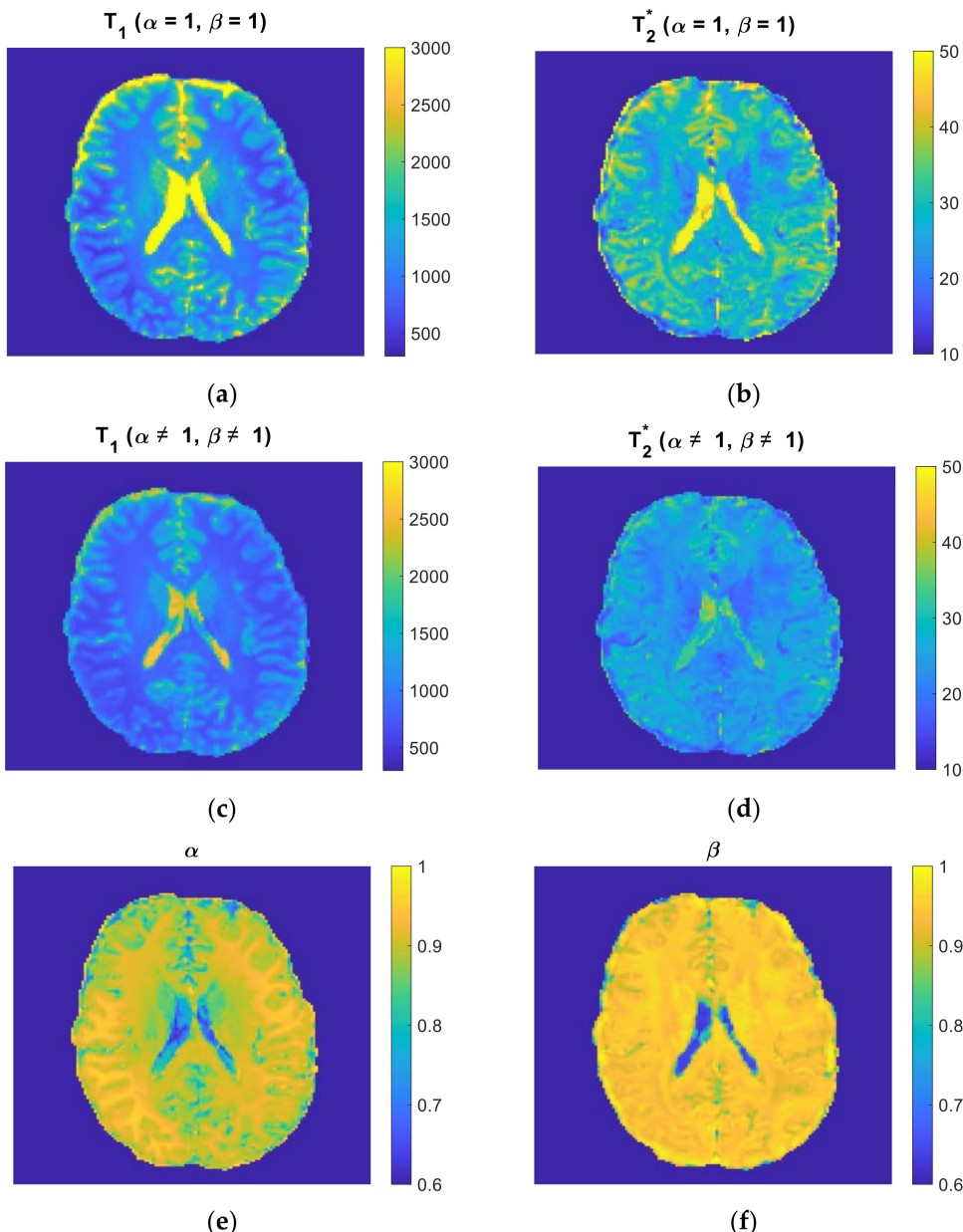

**Figure 6.** An example of spatially resolved maps of model parameters (slice 35, participant 2). Displayed are (**a**) $T_1$ and (**b**) $T_2^*$ for the integer order model. Additionally, time fractional order parameter model parameters (**c**) $T_1$, (**d**) $T_2^*$, (**e**) $\alpha$ and (**f**) $\beta$ are depicted.

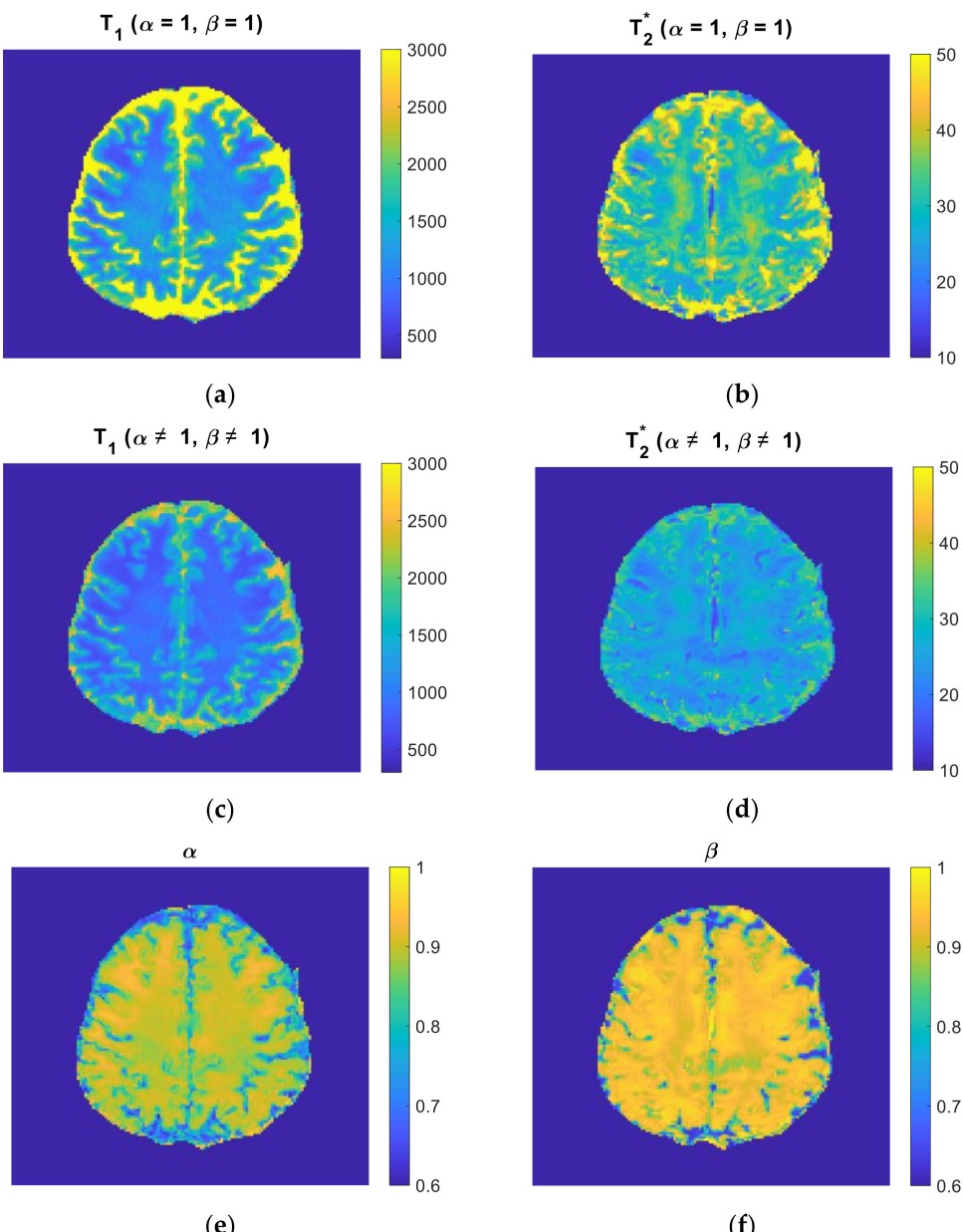

**Figure 7.** A second example of spatially resolved maps of model parameters (slice 50, participant 4). Displayed are (**a**) $T_1$ and (**b**) $T_2^*$ for the integer order model. Additionally, time fractional order parameter model parameters (**c**) $T_1$, (**d**) $T_2^*$, (**e**) $\alpha$ and (**f**) $\beta$ are depicted.

Table 1 provides the improvement in the results of matching the MRF simulated signal with the acquired MRI signal based on the integer order Bloch equations and time fractional order counterpart. It has previously been demonstrated that $T_1$ and $T_2^*$ are not selective to cortical regions in a manner full cortical parcellation of the human cerebral cortex can be achieved in individuals. We also found that $T_1$ and $T_2^*$ mapped using the MRF protocol are unable to differentiate the cortical regions identified in Section 2.6. However, we did find both mapping of $\alpha$ and $\beta$ provides specificity to different cortical regions. Figure 8 are violin plots for $\alpha$ and $\beta$ over the various regions categorised into four groupings: somatosensory cortical region (BA1, BA2, BA3a and BA3b), the primary motor (BA4a and BA4p) and pre-motor (BA6) cortical region, visual cortex region (BA17 and BA18) and two adjacent Broca cortical regions (BA44 and BA45). Based on the notch plot, $\alpha$ was found to be significantly different in the somatosensory cortical region and the primary and pre-motor cortical region. Interestingly, $\beta$ was significantly different in the sub-regions of the four

cortical areas studied. This finding is very interesting, as the differentiation of the Broca regions are usually among the most challenging parcellation tasks in neuroscience.

In Figure 9, the relationships between time fractional Bloch equations parameters are investigated in the entire human brain. We performed tests when $\Delta f$ was a fitted parameter. In this case, see Figure 9a, $\alpha$ and $\beta$ values take on wide ranging values and $\beta$ appears to be linked with $\Delta f$, Figure 9b. By setting $\Delta f = 0$, a vastly different relationship between $\alpha$ and $\beta$ is presented (compare Figure 9a,c). A systematic trend between $\alpha$ and $\beta$ when $\Delta f = 0$ cannot be discerned.

**Table 1.** Improvement in MRF signal matching using the time fractional instead of the integer order Bloch model. Displayed are the root-mean-squared error (RMSE) reduction and corresponding improvement in the dot product computed angle, $\phi$. The t-test *p*-values testing whether the RMSE and $\phi$ improvements are significantly greater than zero have been provided.

| Metric | RMSE Reduction (%) | $\phi$ |
|---|---|---|
| Median | 96.5 | 98.2 |
| Mean | 92.6 | 96.1 |
| Standard deviation | 9.6 | 5.4 |
| *p*-value | $<10^{-9}$ | $<10^{-9}$ |

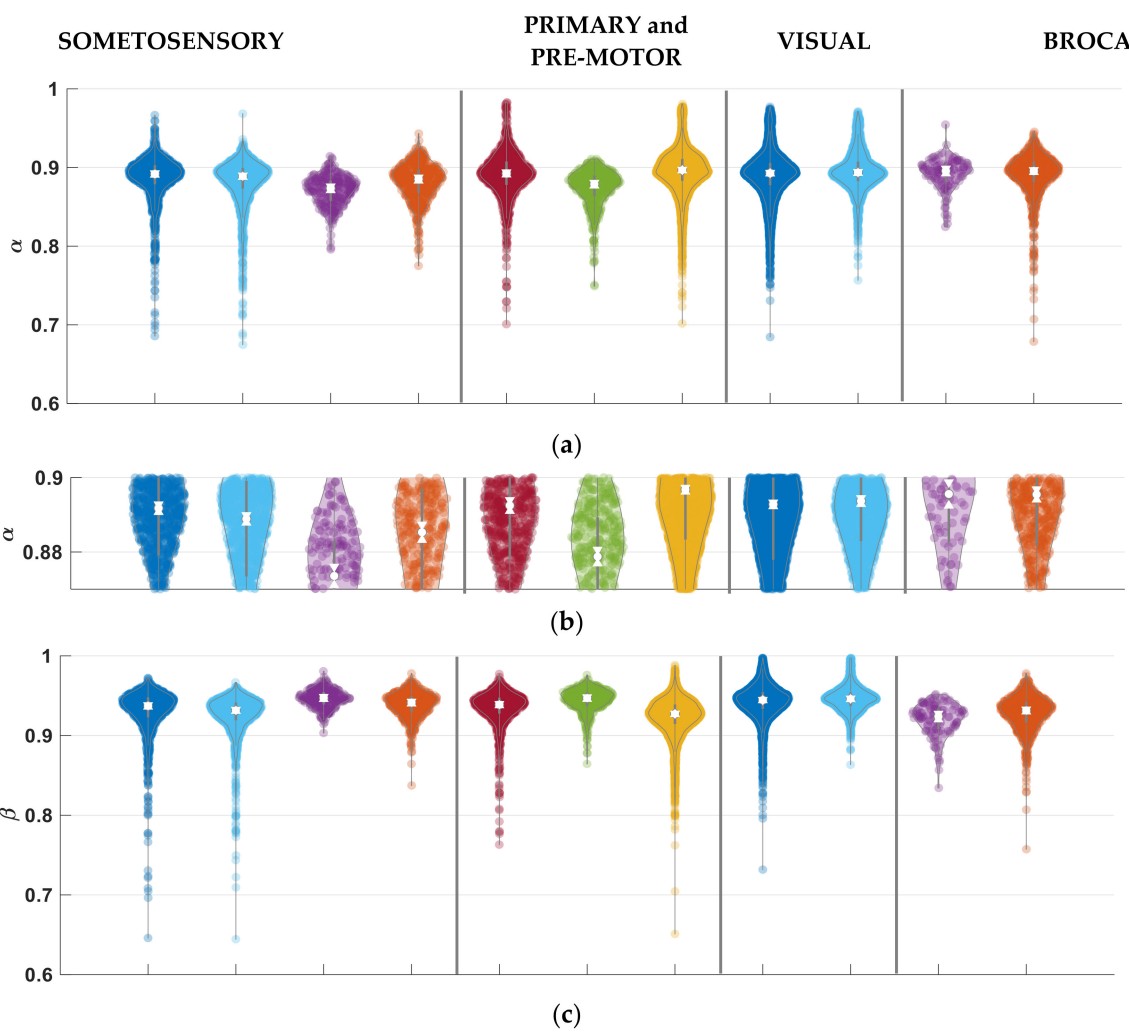

**Figure 8.** *Cont.*

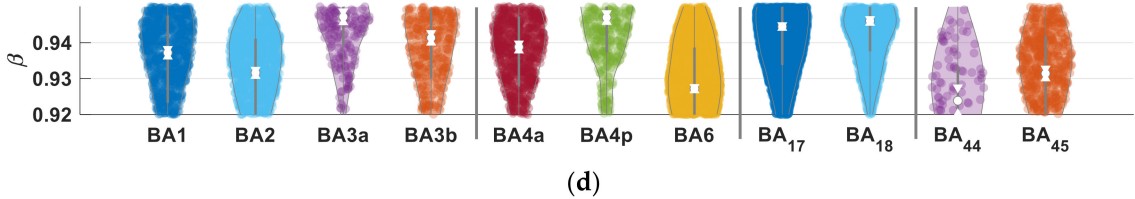

(**d**)

**Figure 8.** Violin plots for $\alpha$ and $\beta$ including notches at the ($p < 0.05$) significance level across the 11 cortical regions considered. The headings identify different cortical areas consisting of different regions, for example, the adjacent cortical areas BA17 and BA18 in the visual cortex have been evaluated. The time fractional Bloch model parameter (**a,b**) $\alpha$ was not able to discern regions ($p < 0.05$) in the visual and Broca cortical regions, whereas (**c,d**) $\beta$ had significant differences between sub-regions in the four cortical regions evaluated.

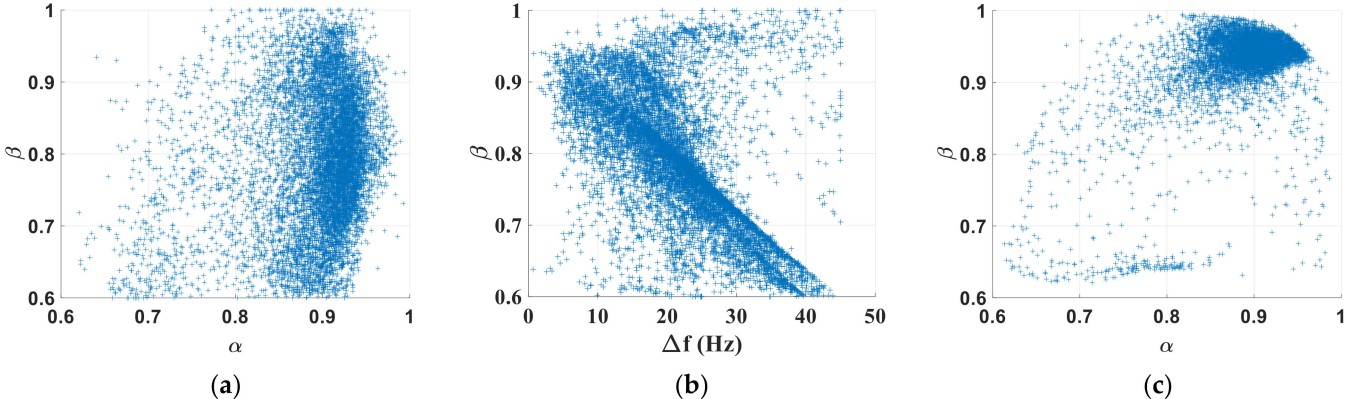

(**a**)             (**b**)             (**c**)

**Figure 9.** Scatter plots to assess time fractional Bloch equations parameter dependency in the human brain. Displayed are (**a**) the relationships between $\alpha$ and $\beta$ for the case when $\Delta f \neq 0$ and (**b**) between $\Delta f$ and $\beta$. When $\Delta f = 0$ is assumed, the (**c**) $\alpha - \beta$ relationship results. The $\Delta f - \beta$ trend may be perceived as linear, with a negative slope.

## 4. Discussion

The aim of the research was to establish the utility of the time-fractional Bloch equations in magnetic resonance fingerprinting for brain imaging studies. The classical integer orders in time Bloch equations are known to produce substantial residuals in the MRF fit and residuals contain information not explained by classical model parameters [54]. The incorporation of time-fractional exponents resulted in a reduction in the fitting error (see Table 1). However, a dependency between fractional exponent $\beta$ and $\Delta f$ was found, resulting in overfitting (see Figures 2 and 3). When $\Delta f = 0$ in the time fractional Bloch equations, overfitting does not occur and exquisite spatially resolved maps of $\alpha$ and $\beta$ can be obtained (see Figures 6 and 7). This approach allowed us to separate out different cortical regions in the brain, with $\beta$ providing better results than $\alpha$ (see Figure 8).

### 4.1. MRF Parameter Discretisation and Matching

Another study considered time-fractional order Bloch equations in an MRF implementation [20]. The method was applied in a $T_1/T_2$ phantom and, using the time-fractional model, better estimates of $T_1$ and $T_2$ were generated in comparison with the integer order Bloch equations implementation. Notably, the authors considered the case of $T_2$ relaxation, i.e., no signal dephasing produced based on the spin-echo acquisition protocol.

The parameters were discretised using the following choices: $T_1$ values in the range (100, 4500) in steps of 100 ms; $T_2$ values in the range (10, 1000) in steps of 10 ms; both $\alpha$ and $\beta$ were in the range (0.96, 1.1) in steps of 0.01. The number of MRF repetitions was 600. It was found that $T_1$ and $T_2$ estimates improved when $\alpha$ and $\beta$ were free parameters and it was additionally reported that $\alpha > \beta$.

The MRF dictionary contains a set of discrete parameters and the finer the increments the larger the time required to obtain a match. As such, there is a trade-off between dictionary size and the computational resource requirements to make a signal match. We chose to have a coarse-grained approach: $T_1$ and $T_2^*$ increments of around 1.5–4%, whereas $\alpha$, $\beta$ and $\Delta f$ as much as 10% increments. Results presented in Figure 4 suggest that for a 10% change in $T_1$, one can expect around 1.6% change in matched angle (assuming $90^\circ$ is the maximum). For $T_2^*$ at the same level of change, we expect about half as much change in the dot product produced angle between MRF simulation and MRI data acquisition over repetitions. Interestingly, $\alpha$ has the largest influence on the angle, which is about double that of $T_1$ and $\beta$ is similar to $T_1$. Based on these findings, in the future consideration should be made on how the different model parameters are discretised. Ideally, the steps chosen for each parameter should approximately produce the same change in the angle between the MRF repetition-based acquired and simulated signals.

*4.2. Role of $\alpha$ and $\beta$*

In the classical case $\alpha = \beta = 1$, which reduces the time-fractional Bloch equations to their integer order form. We opted to perform the study using distinct values for $\alpha$ and $\beta$, as it has been shown that these values are unlikely to be the same [20,37,38]. These previous studies have suggested $\alpha > \beta$ and additionally it was proposed that $\alpha = 1$, resulting in mono-exponential recovery of magnetisation to $M_0$ [37,38]. Interestingly, our brain imaging findings suggest that $\alpha > \beta$ does not hold in the brain, refer to Figures 6 and 7, and also $\alpha$ should not be set to 1. Out of interest, we provided sensitivity results for $\beta$ when $\alpha = 1$, see Figure 3f, and found that this choice does not provide a benefit for model fitting.

It is well established that both $T_1$ and $T_2$ have a level of frequency reliance, which depends on the underlying composition of the material imaged and likely scales as a power law [55]. This effect is particularly pronounced in the presence of lipid/protein structures, where dipole–dipole coupling frequents [56]. The brain consists of large proportions of proteins and lipids, myelin being an example of highly concentrated and organised lipid structures in the brain. It is also known that lipids generally have magnetic properties which influence the MRI magnetic field responsible for the net magnetisation, $M_0$. Our human brain results in Figure 9 showed a potential relationship between $\beta$ and $\Delta f$, the latter being the field change induced frequency shift in the MRI signal. We found that the use of both parameters resulted in overfitting (Figure 3 versus Figure 4), suggesting that one of these two parameters is sufficient in explaining the trend in the MRI signal. Simulation findings presented in Figures 3 and 4 and subsequently in the human brain, see Figure 9, imply that $\beta$ and $\Delta f$ vary together. In view of previous findings on $T_1$ and $T_2$ frequency dependence and noting that the difference between $T_2$ and $T_2^*$ is attributed to tissue induced magnetic field inhomogeneities, our results, interestingly, suggest that $\beta$ captures information on induced field change. That is, an increase in $\Delta f$ corresponds with a decrease in $\beta$, particularly above $\Delta f = 20$ Hz, and remains to be further evaluated for small $\Delta f$ values. Ideally, an analytic expression linking the two would provide the best insight into the relationship between these two parameters. Figure 9 results additionally suggest that $\alpha = \beta$ is unlikely and setting $\alpha = 1$ does not seem appropriate for brain studies. Whilst in this work we have not attempted to demonstrate a clear relationship between $T_2$ and $T_2^*$, it would be interesting to find $T_2^* = T_2^\beta$. This is purely an observation based on $T_2$ and $T_2^*$ values reported for the brain and future work would need to investigate carefully whether such a relationship is mathematically plausible.

*4.3. Cortical Parcellation*

Parcellation of the human cerebral cortex in individuals is challenging and a robust non-invasive imaging method of achieving this goal has not been proposed to date. The time-fractional Bloch equations appears to provide good sensitivity to different cortical regions through model parameters, as illustrated in Table 1. In our study the time-fractional exponent of the transverse components of the magnetic field, $\beta$, resulted in better cortical

area differentiation than $\alpha$, with the time-fractional exponent influencing longitudinal magnetisation evolution. Previously, it was shown that the integer order Bloch equations within an MRF framework result in signal residuals depicting trends useful in delineating cortical regions of distinct cyto-architectures and myelo-architectures [54]. The hypothesis that the cyto-architectures of cortical regions results in induced magnetic field changes distinct to regions appears valid based on the specificity of $\beta$, which is a plausible proxy for cortical tissue induced field changes via $\Delta f$.

## 5. Conclusions

Mathematical model-based approaches of extracting information from MRI data provide an important avenue for producing quantitative parametric maps, reflecting tissue properties. During the last two decades, MRI signal models based on fractional calculus have had growing success in the description of complex biological microstructure associated with temporal memory and spatial heterogeneity. In this paper, our time-fractional order Bloch equations magnetic resonance fingerprinting results suggest that non-integer order approaches can result in a better explanation of the trends in the magnetic resonance fingerprinting signal. Moreover, the time-fractional exponents, one associated with magnetisation recovery and the other with transverse magnetisation loss, can provide new insights in human brain studies involving tissue specific parameter estimations. Our example application involved the parcellation of the human brain using time-fractional exponents of the Bloch equations. This work suggests the utility of fractional-order models to describe magnetic resonance fingerprinting signals in biological tissues.

**Author Contributions:** Conceptualisation, V.V. and S.M.; methodology, V.V. and S.M.; software, S.M. and V.V.; validation, S.M., Q.Y. and D.C.R.; formal analysis, V.V.; investigation, V.V.; resources, V.V. and D.C.R.; data curation, S.M.; writing—original draft preparation, V.V.; writing—review and editing, V.V., S.M., Q.Y. and D.C.R.; visualisation, V.V. and S.M.; supervision, D.C.R. and V.V.; project administration, V.V.; funding acquisition, V.V., Q.Y. and D.C.R. All authors have read and agreed to the published version of the manuscript.

**Funding:** This research was funded by an Australian Research Council Discovery Project Grant (chief investigators include V.V. and Q.Y.), grant number DP190101889. S.M. holds a post-doctoral position as part of the Australian Research Council Training Centre for Innovation in Biomedical Imaging Technology (chief investigators include D.C.R. and V.V.), grant number IC170100035. Q.Y. holds an Australian Research Council Discovery Early Career Research Award, grant number DE150101842.

**Institutional Review Board Statement:** The 7T MRI data collection protocol was reviewed and approved by the University of Queensland Huma Ethics Committee (2005000502).

**Informed Consent Statement:** Written informed consent was obtained from all participants involved in the study.

**Data Availability Statement:** Data generated for this study may be requested by contacting the corresponding author.

**Acknowledgments:** We thank the participants involved in this study as well as Nicole Atcheson and Aiman Al-Najjar for their help with acquiring the data. The authors acknowledge the facilities and the scientific and technical assistance of the National Imaging Facility, a National Collaborative Research Infrastructure Strategy (NCRIS) capability, at the Centre for Advanced Imaging, University of Queensland.

**Conflicts of Interest:** The authors declare no conflict of interest.

## Appendix A

We now derive the relationships $\left(\frac{t}{T_2 \tau_2^{\beta-1}}\right)^{\beta} \geq \left(\frac{t}{T_1 \tau_1^{\alpha-1}}\right)^{\alpha}$ for the time fractional order MRI relaxation process, which reduces to $T_1 \geq T_2$ when $\alpha = \beta = 1$. To illustrate the process involved, we start with the (i) integer case and then provide the (ii) general form.

(i) Let us consider the simplest case for the solutions to the integer order Bloch equation, i.e., (4), by setting $M_z(0) = 0$, $M_x(0) = M_0$, $M_y(0) = 0$ and $\Delta\omega = 0$. Then the solutions can be simplified as follows:

$$M_z(t) = M_0\left(1 - e^{-\frac{t}{T_1}}\right) \tag{A1}$$

$$M_{xy}(t) = M_0 e^{-\frac{t}{T_2}} \tag{A2}$$

where $M_{xy}(t)$ denotes magnetisation in the transverse plane. Since total magnetisation is defined by $M_0$, the following is obtained.

$$\left|M_z(t)\right|^2 + \left|M_{xy}(t)\right|^2 \le \left|M_0\right|^2. \tag{A3}$$

Substituting (A1) and (A2) into (A3) provides us with the following.

$$\left(1 - e^{-\frac{t}{T_1}}\right)^2 + e^{-\frac{2t}{T_2}} \le 1 \tag{A4}$$

Letting $x' = e^{-\frac{t}{T_1}}$ and $y' = e^{-\frac{t}{T_2}}$, (A4) becomes the following.

$$(1 - x')^2 + y'^2 \le 1. \tag{A5}$$

As $t \in [0, +\infty)$, $x' \in [0, 1]$ and $y' \in [0, 1]$. The relationship (A5) is depicted in Figure A1. The area under the thick red curve represents all the $x'$ and $y'$ values satisfying (A5). In the blue shaded area, $y' \le x'$, i.e., $e^{-\frac{t}{T_2}} \le e^{-\frac{t}{T_1}}$, and hence $T_1 \ge T_2$.

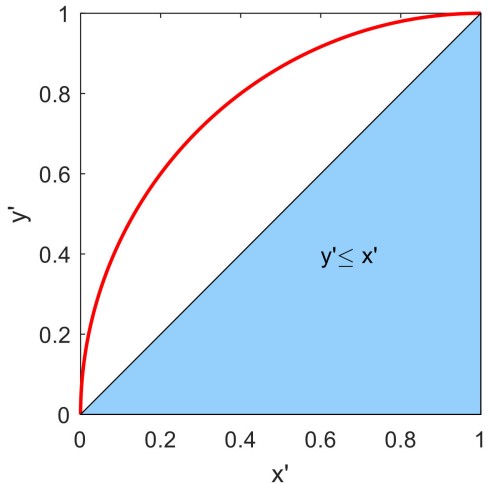

**Figure A1.** Illustration of the relationship in (A5). The area under the red curve represents all the $x'$ and $y'$ values that satisfy (A5). The blue shaded area corresponds with $y' \le x'$, implying $T_1 \ge T_2$.

(ii) We now consider the simplest case for the solutions to the time fractional order Bloch equation, (3), by setting $M_z(0) = 0$, $M_x(0) = M_0$, $M_y(0) = 0$ and $\Delta\omega = 0$ in the following.

$$M_z(t) = \frac{M_0}{T_1} \tau_1^{1-\alpha} t^\alpha E_{\alpha,\alpha+1}\left(-\frac{\tau_1^{1-\alpha} t^\alpha}{T_1}\right) = M_0\left(1 - E_\alpha\left(-\frac{\tau_1^{1-\alpha} t^\alpha}{T_1}\right)\right), \tag{A6}$$

$$M_{xy}(t) = M_0 E_\beta\left(-\frac{\tau_2^{1-\beta} t^\beta}{T_2}\right) \tag{A7}$$

Substituting (A6) and (A7) into (A3) gives the following.

$$\left(1 - E_\alpha\left(-\frac{\tau_1^{1-\alpha}t^\alpha}{T_1}\right)\right)^2 + \left(E_\beta\left(-\frac{\tau_2^{1-\beta}t^\beta}{T_2}\right)\right)^2 \leq 1 \tag{A8}$$

.

Letting $x' = E_\alpha\left(-\frac{t^\alpha}{T_1\tau_1^{\alpha-1}}\right)$ and $y' = E_\beta\left(-\frac{t^\beta}{T_2\tau_2^{\beta-1}}\right)$, (A8) becomes $\left(1-x'\right)^2 + y'^2 \leq 1$, which is the same as in (A5). Again, as $t \in [0, +\infty)$, $x' \in [0,1]$ and $y' \in [0,1]$ for $y' \leq x'$ we have $E_\beta\left(-\frac{t^\beta}{T_2\tau_2^{\beta-1}}\right) \leq E_\alpha\left(-\frac{t^\alpha}{T_1\tau_1^{\alpha-1}}\right)$, i.e., $\left(\frac{t}{T_2\tau_2^{\beta-1}}\right)^\beta \geq \left(\frac{t}{T_1\tau_1^{\alpha-1}}\right)^\alpha$, which reduces to the classical case when $\alpha = \beta = 1$.

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
