# Peer review of "Fractional Order Magnetic Resonance Fingerprinting in the Human Cerebral Cortex"

_mathematics, doi:10.3390/math9131549_

Round 1
Reviewer 1 Report
The manuscript is really well presented and the introduction provides a good overview of MRF. The results could be slightly more clear with additional and improved figures to show the parcellation sensitivity.
Line 230: There are some minor grammatical errors that I won’t point out (ex. missing commas)
Line 239: a person’s initials need not be included.
Line 245: Can you comment on the use of GRAPPA and if this has an effect on MRF sensitivity?
Line 293: include Mz(t) “shown in red”
Line 294: the R^2 looks a little off, no lines for insignificant lines of best fit? Unclear
Figure placements are not aligned with where they are mentioned in this text.
Figure 9 x-axis not visible in first row of graphs, including notches at the significance level I can’t see? As differentiating regions is an important finding I’d like to be able to see this (perhaps with an additional figure?)
Reviewer 2 Report
- The abstract and conclusions should better explain fractional calculus, motivating why these is necessary for this particular application. And these explanations should be accompanied of comments on the greater applicability of fractional calculus.
- The shortcomings of fractional calculus should be highlighted in the paper. Such as the greater computational complexity. How is the long memory effect handled in the paper?
- Why the Caputo definition was chosen for the paper? (as opposed to the other fractional derivative definitions).
- Overall it has to be better justified why the proposed approach was chosen in the first place (prior to testing it).
- In Fig 8 it is not clear how many outliers are there (in red). Also, why so many outliers? It would be convenient to add to the figure the position of all the points (RMSR) besides the boxplot. Further, please specify the size of the whiskers.
Round 2
Reviewer 2 Report
I do not have further comments.